# An Integrated Mass Spectrometry-Based Glycomics-Driven Glycoproteomics Analytical Platform to Functionally Characterize Glycosylation Inhibitors

**DOI:** 10.3390/molecules27123834

**Published:** 2022-06-14

**Authors:** Michael Russelle S. Alvarez, Qingwen Zhou, Sheryl Joyce B. Grijaldo, Carlito B. Lebrilla, Ruel C. Nacario, Francisco M. Heralde, Jomar F. Rabajante, Gladys C. Completo

**Affiliations:** 1Institute of Chemistry, College of Arts and Sciences, University of the Philippines Los Baños, Los Baños 4031, Philippines; mralvarez@ucdavis.edu (M.R.S.A.); sbgrijaldo@up.edu.ph (S.J.B.G.); rcnacario@up.edu.ph (R.C.N.); 2Department of Chemistry, University of California Davis, Davis, CA 95616, USA; qwzzhou@ucdavis.edu (Q.Z.); cblebrilla@ucdavis.edu (C.B.L.); 3Lung Center of the Philippines, Quezon City 1100, Philippines; fmheralde1@up.edu.ph; 4Institute of Mathematical Sciences and Physics, College of Arts and Sciences, University of the Philippines Los Baños, Los Baños 4031, Philippines; jfrabajante@up.edu.ph

**Keywords:** glycomics, glycoproteomics, glycosylation, proteomics, in silico docking, network pharmacology, non-small cell lung cancer

## Abstract

Cancer progression is linked to aberrant protein glycosylation due to the overexpression of several glycosylation enzymes. These enzymes are underexploited as potential anticancer drug targets and the development of rapid-screening methods and identification of glycosylation inhibitors are highly sought. An integrated bioinformatics and mass spectrometry-based glycomics-driven glycoproteomics analysis pipeline was performed to identify an N-glycan inhibitor against lung cancer cells. Combined network pharmacology and in silico screening approaches were used to identify a potential inhibitor, pictilisib, against several glycosylation-related proteins, such as Alpha1-6FucT, GlcNAcT-V, and Alpha2,6-ST-I. A glycomics assay of lung cancer cells treated with pictilisib showed a significant reduction in the fucosylation and sialylation of N-glycans, with an increase in high mannose-type glycans. Proteomics analysis and in vitro assays also showed significant upregulation of the proteins involved in apoptosis and cell adhesion, and the downregulation of proteins involved in cell cycle regulation, mRNA processing, and protein translation. Site-specific glycoproteomics analysis further showed that glycoproteins with reduced fucosylation and sialylation were involved in apoptosis, cell adhesion, DNA damage repair, and chemical response processes. To determine how the alterations in N-glycosylation impact glycoprotein dynamics, modeling of changes in glycan interactions of the ITGA5–ITGB1 (Integrin alpha 5-Integrin beta-1) complex revealed specific glycosites at the interface of these proteins that, when highly fucosylated and sialylated, such as in untreated A549 cells, form greater hydrogen bonding interactions compared to the high mannose-types in pictilisib-treated A549 cells. This study highlights the use of mass spectrometry to identify a potential glycosylation inhibitor and assessment of its impact on cell surface glycoprotein abundance and protein–protein interaction.

## 1. Introduction

Lung cancer is the leading cause of cancer-related mortalities worldwide [1,2]. Cancer incidence and mortality are increasing worldwide, reflecting several factors: aging; population growth; cancer risk factors; and socioeconomic development. According to the GLOBOCAN 2020 database, there will be 19.3 million new cases and 10 million cancer deaths worldwide [3]. Out of these, 2,206,771 cases (11.4%) and 1,796,144 (18.0%) deaths for both sexes will be due to lung cancer.

Protein glycosylation is one of the most complex and most frequent post-translational modifications and is involved in many of the cellular interactions, such as host–pathogen interactions, cell differentiation and trafficking, and intra- and intercellular signaling [4,5,6]. Protein glycosylation is a complex process that starts at the endoplasmic reticulum and is continued in the Golgi apparatus, where the glycans are further processed to achieve the diversity and complexity of the final glycan structures, through a series of steps involving glycosyltransferases and glycosidases [7,8]. The overexpression of these glycan-processing enzymes is usually observed in cancer cells, resulting in an enhanced expression of the related glycan structures. For example, the enzymes Alpha1-6FucT, B4GALT2, MAN1A2, and MAN2A1 are overexpressed in lung cancer tissue samples [9]. Likewise, glycans synthesized by these enzymes are also overexpressed in lung cancer tissues [10,11]. Additionally, aberrant glycosylation leads to increased biosynthesis of various tumor antigens, such as Sialyl Lewis X (sLe^x^), which serves as a ligand for the cell adhesion molecule, selectin. This antigen is also involved in the adhesion of cancer cells to the vascular endothelium and hematogenous metastasis. Cancer progression is also associated with changes in the glycosylation of cell-surface proteins involved in the loss of cell-to-cell adhesion and increased metastatic potential. Furthermore, altered glycosylation is also correlated with the other hallmarks of cancer, such as enhanced proliferation, angiogenesis potential, replicative immortality, metastatic potential, apoptosis, and tumor suppression [12,13].

Several glycosyltransferases have been associated as cancer biomarkers [4,9,11]. A glycosyltransferase used as a biomarker is UDP-N-acetyl-D-glucosamine: N-acetylglucosamine transferase V (GlcNAcT-V), which catalyzes the β1-6 branching of N-glycans. Increased β1-6 branching, due to GlcNAcT-V overexpression, has been observed in breast carcinoma [14]. Sialyltransferases are glycosyltransferases that are abnormally expressed in cancers and are involved in carcinogenesis, progression, and metastasis [15,16,17]. An overexpression of α2-3 sialyltransferase III (ST3Gal-III) in pancreatic cancer has been implicated in pancreatic tumor progression. The overexpression of α2-6 sialyltransferase I (ST6GalNAc-I) was related to poor patient survival in colorectal carcinoma patients [18]. As such, the glycosyltransferases and glycosidases are underexploited drug targets for cancer therapeutics and there is a relative lack of small molecule inhibitors of these enzymes with drug-like properties. The glycosylation inhibitors that were previously reported include metabolic inhibitors, which target the formation of nucleotide sugars; tunicamycin which targets dolichol-PP-GlcNAc formation (biosynthesis of N-glycans); plant alkaloids that inhibit the processing of glycosidases; substrate analogs which are specific towards glycosyltransferases and glycosidases; glycoside primers which divert the assembly of glycans from endogenous acceptors towards exogenous primers; and tagged monosaccharides which target several different biosynthesis pathways [6]. Synthetic compounds, such as 2-deoxy-2-fluorofucose and 2,4,7,8,9-Penta-O-acetyl-N-acetyl-3-fluoro-b-D-neuraminic acid ester, were previously shown to inhibit the expression of fucosylated and sialylated N-glycan structures, respectively, in the glycocalyx of Caco-2, A549, and PNT2 cells, using LC–MS/MS methods [19]. The natural product, tunicamycin, induces the inhibition of the protein N-glycosylation by blocking the GlcNAc phosphotransferase-catalyzed transfer of N-acetylglucosamine-1-phosphate from UDP-GlcNAc to dolichol-P, which results in the decreased production of dolichol-PP-GlcNAc. In combination with anticancer drugs, tunicamycin has also been shown to be cytotoxic against multidrug-resistant human ovarian cystadenocarcinoma cells, by inhibiting protein and glycoprotein syntheses [20]. However, the effects of these glycan inhibitors on cancer-associated pathways, and in correlation with protein glycosylation, have not been explored before.

In this study, we integrated LC–MS/MS methods—glycomics, proteomics, and glycoproteomics—to characterize the effect on protein glycosylation of a small-molecule inhibitor, pictilisib, identified through our computational docking predictions. In conjunction with the LC–MS/MS methods, we developed molecular and bioinformatics models to understand how pictilisib affects the protein glycosylation, and subsequently affects the cancer-associated biological pathways.

## 2. Results

### 2.1. Pictilisib Was Validated to Reduce the Relative Abundance of Fucosylated and Sialylated N-glycans

We previously discovered, through in silico screening, that pictilisib is able to bind and inhibit several glycosyltransferases [21]. To determine the effect of changes in the protein glycosylation after pictilisib treatment, in vitro assays were performed using an A549 non-small cell lung cancer (NSCLC) cell line as a model for lung cancer. Dose–response cytotoxic assay and preliminary drug-titration assay after 24 h of treatment were conducted to determine the nontoxic drug concentrations that could still affect the protein glycosylation (Appendix A). After optimization of the assay conditions, the A549 cells were treated with pictilisib (4 µM) for 24 h, then subjected to glycomics profiling using an established mass spectrometric method [22]. Glycomics profiling with mass spectrometry allows for a comprehensive and reproducible analysis of the glycan composition of the cell’s glycocalyx, after treatment with pictilisib or vehicle control (Figure 1a and Appendix A). Comparing the sum of the relative abundances of the primary N-glycan types—high-mannose, undecorated, fucosylated, sialylated, and sialo-fucosylated—shows that pictilisib treatment significantly reduced the total relative abundances of the fucosylated and sialylated N-glycans (Figure 1b). A total of 138 glycans were profiled, of which 36 were found to be significant (*p* < 0.05). A closer inspection of these N-glycans showed that the total fucosylated complex- and high-mannose-type N-glycans, as well as both the total sialylated complex- and hybrid-type N-glycans were significantly reduced by the pictilisib treatment (Figure 1b). Specifically, the N-glycan compositions Hex_6_HexNAc_4_NeuAc_1_, Hex_6_HexNAc_4_NeuAc_2_, Hex_3_HexNAc_2_Fuc_1_, Hex_5_HexNAc_4_Fuc_2_, Hex_7_HexNAc_6_Fuc_1_, Hex_8_HexNAc_7_Fuc_6_, Hex_8_HexNAc_7_Sia_2_, Hex_9_HexNAc_8_Fuc_1_, and Hex_9_HexNAc_8_Fuc_1_NeuAc_2_ were found to be very significantly underexpressed in the pictilisib-treated cells (Table 1). Mapping these N-glycan compositions to the known N-glycan biosynthetic pathway shows potential glycosylation enzyme reactions that could be inhibited from the pictilisib treatment (Figure 1c), specifically those glycosylation reactions involving the addition of fucose and sialic acid residues [23,24]. These significantly underexpressed N-glycans represent several known cancer-related N-glycan epitopes, such as Lewis and Sialyl Lewis antigens, core fucosylation, and α2,6-sialylated lactosamine [13].

### 2.2. Proteomic Analysis Shows Upregulated Pathways Involving Apoptosis and Cell Adhesion, and Downregulated Pathways Involving Cell Cycle Process, mRNA Processing, and Protein Translation

To validate the bioactivity effects of pictilisib on A549, we conducted in vitro assays coupled with label-free quantitative proteomics, to identify the specific pathways targeted by pictilisib. Proteins were filtered by setting the Byologic score higher than or equal to 100 and having two unique peptides per protein. The protein intensities were reported as the sum of the top two peptides for each protein, normalized to the total intensity per sample. The dataset was further filtered based on the presence of a specific protein in at least two replicates per group and then analyzed using multiple *t*-tests (α = 0.05) (Appendix A). Based on the proteomics data (Appendix A), 1518 proteins were quantified, with 380 proteins identified as significantly different (*p*-value < 0.05, Figure 2a). A gene set enrichment analysis of these significantly different proteins showed interesting biological processes affected by the pictilisib treatment, such as the upregulation of apoptosis and biological adhesion processes and the downregulation of cell cycle processes (Figure 2b,c, Appendix A). In vitro apoptosis and cell cycle assays verify that, indeed, pictilisib treatment induced apoptosis (Figure 2d and Appendix A) and G0/G1 cell cycle arrest (Figure 2e and Appendix A) in A549 cells. Correspondingly, the quantification of specific apoptosis, cell cycle, and DNA damage-related proteins show significant differences in key proteins, such as HELLS [25], TOP2A [26,27], MCM6 [28], PSMB4 [29], EIF4G1 [30,31,32], EIF5A [33], DDX5 [34], and RACK1 [35,36], involved in these pathways (Figure 2f–h). Likewise, the effect of pictilisib on cell migration was verified using both scratch assay and transwell migration assay, with pictilisib treatment causing a significant reduction in cell migration (Figure 2i,j, Appendix A). Proteomics analysis shows that the mechanism affecting the cell migration was observed by overexpressing adhesion proteins and upregulating cell adhesion pathways (Figure 2k). Interestingly, the pictilisib treatment also significantly downregulated the proteins involved in mRNA processing (Figure 3a,b), and the protein translation (Figure 3c) processes.

### 2.3. Glycoproteins with Reduced Fucosylation and Sialylation Were Involved in Apoptosis, DNA Damage Repair, and Cell Adhesion

Aberrant glycosylation has been well-documented in cancer, with fundamental changes in the glycosylation patterns of cell surface and secreted proteins during cancer progression. Growing evidence supported the role of glycosylation during multiple steps in tumor progression, cancer cell proliferation, invasion, metastasis, and angiogenesis [5,11,13,37,38].

Our glycomics results show that pictilisib treatment significantly reduced global fucosylation and sialylation of the cell membrane N-glycans. Likewise, our proteomics results show that pictilisib treatment significantly affected adhesion, apoptotic, and cell cycle pathways. To identify which of the glycoproteins have reduced fucosylation and sialylation and their involvement in these pathways, we performed quantitative site-specific glycoproteomics coupled with gene ontology analysis of the pictilisib-treated cells. Glycoforms were identified after score-filtering, replicate-filtering (present in at least two of the replicates), and normalized glycopeptides per protein glycosite (Figure 4a; Appendix A). Normalized glycoforms were categorized based on the N-glycan type—high-mannose, undecorated, fucosylated, sialylated, and sialofucosylated—and then summed up for each glycosite. For example, the changes in glycoform occupancy in ANPEP, ADA10, ITGB1, and ITGA3 following pictilisib treatment reduced the fucosylation and sialylation or sialofucosylation in specific glycosites (Figure 4b). The glycosites with reduced fucosylation, sialylation, and sialofucosylation were represented as heat maps annotated with gene ontologies of the corresponding glycoproteins. Indeed, the glycoproteins associated with biological adhesion and locomotion, and apoptosis, had reduced fucosylation, sialylation, and sialofucosylation (Figure 4c). Increasing sialylation leads to a buildup of negative charges, physically disrupting cell-cell adhesion and promotes detachment through electrostatic repulsion [35]. Overexpression of the enzyme ST6GAL1 and its glycan product, sTn, leads to an increased migration and invasion in carcinoma [39].

Interestingly, the pictilisib treatment also reduced fucosylation, sialylation, and sialofucosylation of glycoproteins involved in chemical stimulus–response (Figure 4c). On the other hand, the glycoproteins involved in the immune system response have only reduced sialylation and sialofucosylation after pictilisib treatment. Looking specifically at the pathway effects, such as the integrin pathway, the integrins showed reduced fucosylation, sialylation, and sialofucosylation of several of their glycosites. While the epidermal growth factor receptor (EGFR) pathway and the ubiquitin-proteasome pathway glycoproteins had reduced sialofucosylation following pictilisib treatment (Appendix A), these glycoproteins are also shown to perform functions in binding, catalysis, regulation, signal transduction, transport, and structural support. When mapped to show the protein–protein interaction network using STRING (Figure 4d), the enrichment analysis further confirms that these glycoproteins are significantly enriched in pathways related to cell adhesion, apoptotic process, and signaling pathways, DNA damage responses, and cellular responses to chemical stimuli (Figure 4e).

Site-specific protein glycoproteomics also allowed us to investigate deeper into the molecular interactions between glycoproteins, such as integrins. Integrin α-5 (ITGA5) and integrin ꞵ-1 (ITGB1) are integrins involved in several biological processes, including cell adhesion and survival. Following the pictilisib treatment of the A549 cells, we found several glycosites in both of the glycoproteins that had either reduced fucosylation, sialylation, or sialofucosylation (Figure 5a,c; Appendix A). This site-specific glycosylation information was overlaid with protein domain annotations from PFAM (http://pfam.xfam.org/, accessed on 15 May 2021) [40], to show how certain glycosites could potentially contribute to the protein interactions (Figure 5b,d). Further analysis through molecular dynamics showed that specific glycosites are significantly affected by the type of glycan decorations (Appendix A). Hex_6_HexNAc_7_Fuc_4_NeuAc_2_ was found to be downregulated, while Hex_7_HexNAc_2_ was upregulated in ITGB1 glycosite Asn269, following pictilisib treatment. Glycosite Asn269 was deemed important, due to its proximity to the interaction interface of ITGB1 and ITGA5. These glycans were modeled into ITGA5 and ITGB1 complex (PDB ID: 3vi4) using CHARMM-GUI [41], then simulated over 45 ns using NAMD [42] (Appendix A). Hydrogen-bonding interactions were then monitored in VMD to show additional residue contacts by Hex_6_HexNAc_7_Fuc_4_NeuAc_2_ in the negative control compared to Hex_7_HexNAc_2_ in the pictilisib-treated cells (Figure 5e).

### 2.4. Pictilisib Was Predicted to Interact and Inhibit Glycosylation Enzymes Using In Silico Docking and Network Pharmacology

Combined network pharmacology and in silico docking approaches were previously used to identify potential interactors with N-glycosylation-related proteins [21]. Several gene–drug interaction databases were surveyed, resulting in 185 predicted glycosylation interactors that were mapped against 356 glycosylation-related proteins and enzymes (Figure 6a; Appendix A). From this set of compounds, pictilisib was selected due to its high degree of interactions (Figure 6b). Specifically, pictilisib was predicted to lower the expression of the glycosyltransferase genes B3GALNT1, B4GALT2, and glycosidase MAN1A1 through interactions with PIK3CA [43].

Additionally, the compounds were screened and docked onto the available crystal structures of three glycosylation proteins—Alpha1-6FucT, Alpha2,6-ST I, and GlcNAcT-V—to predict the binding affinities. Here, pictilisib was predicted to bind to the active sites of Alpha1-6FucT, Alpha2,6-ST I, and GlcNAcT-V with a higher binding affinity than the natural substrate (Figure 6c; Appendix A). Our analysis suggests potential pictilisib interactions with critical amino acid residues in each of the enzyme’s active sites. Against Alpha1-6FucT (Alpha1-6FucT), pictilisib formed pi–cation interactions with Arg365, hydrogen-bonding interactions with His363, and pi–pi T-shaped interactions with Tyr220 (Figure 6d). In a similar docking experiment by Manabe et al., the diphosphate group of GDP-fucose was predicted to form hydrogen bonds with Gly221, Arg365, Ser469, and Gln470 [44]. Additionally, the His363 sidechain and Tyr250 backbone residues were shown to tether the guanosine moiety with hydrogen bonds [45]. Pictilisib also formed hydrogen bonding interactions with Gln235, pi–pi T-shaped interactions with His370, and Van der Waals interactions with Ala363 in Alpha2,6-ST I (Figure 6e). It is noteworthy that a proposed reaction mechanism of Alpha2,6-ST I showed His370 acting as the catalytic base for the deprotonation of the 6′-hydroxyl group of the N-glycan acceptor, leading to a S_N_2 attack at the C2 position of Neu5Ac [46]. Due to these predicted interactions, pictilisib was chosen for further in vitro studies, to determine if it can modulate glycosylation. On the other hand, pictilisib formed hydrogen bonding interactions with Trp401, Asp378, and Leu372 of GlcNAcT-V (Figure 6f). The sulfur atom in pictilisib also formed pi–sulfur interactions with Phe380 and Lys554. The same aromatic amino acid residues, Phe380, and Trp401 also interacted with the sugar acceptor, with Trp401 further restraining the conformation of the α1,6-branch. Most of the top ligands also formed hydrogen bonding interactions with Lys554, a residue with known interaction with the sugar acceptor. The residues—Phe390, Trp401, and Lys554—are also found in the acceptor substrate binding site for MGAT-IX, suggesting that these residues are relevant in acceptor sugar recognition.

## 3. Discussion

Glycans also play a role in regulating the processes that lead to cell death, such as controlling intra- and extracellular pathways that promote the initiation and execution of apoptosis [47]. Cancer cells resist apoptosis through the modification of glycans presented on cell death receptors. Glycosylation can also modulate the function of the death receptors of the extrinsic apoptotic pathway, Fas (CD95), and TNFR1 (tumor necrosis factor receptor 1) [48]. These glycosylations may positively regulate the apoptotic machinery. Galectin-3 binding to β1,6 branched glycans regulate the tumor cell motility by stimulating focal adhesion modeling, FAK and PI3K activation, local F-actin instability, and α5β1 integrin translocation to fibrillar adhesions [49]. Lewis a and Lewis b antigens originated from the mono- or di-fucosyl substitution of type 1 chains, while Lewis x and Lewis y are derived from the mono- or di-fucosyl substitution of type 2 chains. The mono-fucosyl substitution of the α2,3-sialylated type 1 or type 2 chains leads to the formation of Sialyl Lewis a (sLe^a^) and Sialyl Lewis × (sLe^x^), respectively. Core fucosylation is also observed in several cancers [5]. This involves the addition of α1,6-fucose to the innermost GlcNAc residue of N-glycans through Fuc-TVIII (FUT8), and overexpression is additionally observed in several cancers, including lung cancer [50]. In breast cancer, the increased core fucosylation of EGFR was associated with an increased dimerization and phosphorylation, resulting in increased EGFR-mediated signaling, promoting tumor growth [37]. The expression of β-galactoside α2,6-sialyltransferase (ST6Gal1) is altered in several cancers, including colon, stomach, and ovarian [51]. The Ras pathway regulates the transcription and expression of ST6Gal1, and transfectants containing ST6Gal1-expressing cells indicate an increased adhesion to the extracellular matrix molecules in colon [52] and breast cancer [18].

Glycans associated with aberrant fucosylation and sialylation are implicated in cancer progression and metastasis [39,53]. Thus, the inhibition of enzymes involved in the biosynthesis of fucosylated and sialylated glycans has attracted interest as a novel strategy in the development of potential anti-cancer therapeutics. Despite advances made in the development of fucosylation and sialylation inhibitors [54], our use of an in silico, mass spectrometry- and bioinformatics-based approach in the identification of a small molecule inhibitor, pictilisib, serves as an alternative and complementary method for rapid identification of potential glycosyltransferase inhibitors.

We previously discovered, through in silico screening, that pictilisib gave the highest binding affinity, among more than 14,000 compounds and drugs, towards the homology-modeled Alpha1-6FucT at −9.3 kcal/mol [21]. Pictilisib was previously identified as a clinically well-tolerated chemotherapeutic agent that specifically targets the enzyme PI3 kinase among patients with advanced solid tumors, mostly arising from colorectal and breast cancers [55]. To the best of our knowledge, this is the first study to investigate the effect of this chemotherapeutic drug on protein glycosylation, specifically with respect to glycosyltransferase inhibition. However, one aspect of protein glycosylation that was not explored in this study was the possible effects of the drugs on the expression of sugar transporters [7,8], glycosyltransferase retention in the Golgi apparatus [56,57], and the overall integrity of the N-glycan biosynthesis machinery. Future experiments utilizing mass spectrometry to characterize these processes can be developed and incorporated into the glycomics-driven glycoproteomics mass spectrometry workflow, such as this.

## 4. Materials and Methods

### 4.1. Biochemical Assays

#### 4.1.1. Cell Culture

The cell line A549 (CCL-185TM) was obtained from the American Type Culture Collection (ATCC). The A549 cells were grown in RPMI 1640 medium (Gibco), supplemented with 10% fetal bovine serum (FBS) and 1% penicillin-streptomycin (Thermo Scientific, Waltham, MA, USA) in T75 flasks. The media was changed every other day. For each assay, the cells were grown in at least three biological replicates and maintained in a humidified incubator at 37 °C and in an atmosphere of 5% CO_2_.

#### 4.1.2. Dose–Response Assay

The A549 cells were seeded into 96-well plates at 3000 cells/well. The plates were incubated at 37 °C, 5% CO_2_, for 24 h to allow attachment and proliferation. After which, the cells were treated with half-log dilutions of pictilisib (SelleckChem, Houston, TX, USA) and negative control (1% *v*/*v* DMSO). The cells were incubated with the test compounds at 37 °C, 5% CO_2_, for 24 h. The cell viability was detected using CellTiter^96^ AQueous MTS assay reagent (Promega), following the manufacturer’s instructions. The IC_50_ value was calculated using GraphPad Prism (version 9.3.1 for Windows, GraphPad Software, San Diego, CA, USA) using % viability as input values for each log (pictilisib) concentration. The assays were completed in triplicate.

#### 4.1.3. Cell Cycle Assay

The A549 cells were seeded into 100 mm plates. Upon reaching approximately 80% confluency, the cells were treated with 4 µM pictilisib (final concentration), 0.1% *v*/*v* DMSO (negative control), or 100 µM docetaxel (positive control) for 24 h at 37 °C, 5% CO_2_. The cell cycle assay was performed using Cellometer™ PI Cell cycle kit (Nexcelom), according to the manufacturer instructions. The data were acquired using Cellometer Vision CBA (version 5, Nexcelom, Lawrence, MA, USA) using the protocol CBA_Cell Cycle-PI660 nm, with an exposure time of 15,000 ms. The acquired image cytometry data were analyzed using FCS Express 7.0 (version 7.12.0007, De Novo Software, Pasadena, CA, USA). Cell gating was adjusted based on negative control. Assays were completed in triplicate.

#### 4.1.4. Apoptosis Assay

The A549 cells were seeded into 100 mm plates. Upon reaching approximately 80% confluency, the cells were treated with 4 µM pictilisib (final concentration), 0.1% *v*/*v* DMSO (negative control), or 100 µM docetaxel (positive control) for 24 h at 37 °C, 5% CO_2_. The cell cycle assay was performed using Annexin V-FITC/PITM Apoptosis kit (Nexcelom), according to the manufacturer’s instructions. Data were acquired using Cellometer Vision CBA (version 5, Nexcelom, Lawrence, MA, USA), using the protocol CBA_Annexin V + PI assay, with an F1 exposure time of 8000 ms and F2 exposure time of 20,000 ms. Acquired data were analyzed using FCS Express 7.0. Cell gating was adjusted based on negative and positive controls. Assays were completed in triplicate.

#### 4.1.5. Scratch Assay

The A549 cells were seeded into 6-well plates and allowed to grow to confluency at 37 °C, 5% CO_2_. Cell surface scratches were made using P200 pipette tips, then washed twice with PBS to remove the debris. The plates were supplemented with RPMI media (2% FBS, 1% penicillin-streptomycin) to reduce the effects of cell proliferation. The cells were treated with a final concentration of 4 µM pictilisib or with 0.1% *v*/*v* DMSO (negative control) for 48 h. Micrographs were taken starting from 0 h and every 12 h thereafter. The wound size areas were measured using ImageJ software [58] and reported relative to initial wound size. Assays were completed in triplicate.

#### 4.1.6. Transwell Migration Assay

The A549 cells were grown in 100 mm plates until reaching approximately 80% confluency. Then, the cells were treated to a final concentration of 4 µM pictilisib or with 0.1% *v*/*v* DMSO (negative control) for 24 h. After which, the cells were harvested using trypsin and adjusted to 50,000 cells per mL in RPMI media (1% penicillin-streptomycin) without FBS. One (1) mL of the resulting suspension was pipetted into the top compartment of a transwell plate. The bottom chamber was filled with complete media to establish chemotaxis. The plates were subsequently incubated for 3 h at 37 °C, 5% CO_2_, to allow cell migration. The plates were washed twice with HBSS (Hank’s Balanced Salt Solution), fixed with 1% formaldehyde for 5 min, and the bottom compartment stained with 0.1% crystal violet. The cells that migrated through the transwell membrane were visualized in micrographs and manually counted using ImageJ analysis software [58]. Assays were completed in triplicate.

### 4.2. Mass Spectrometry Assay and Data Processing

#### 4.2.1. Cell Treatment and Glycan, Protein, and Glycoprotein Enrichment

The cells were grown in T75 flasks until reaching approximately 80% confluency. The cells were treated to a final concentration of 4 µM pictilisib or with 0.1% *v*/*v* DMSO (negative control) for 24 h. For the glycomic and proteomic mass spectrometric assay, the cells were grown in triplicate T75 flasks for each group (*n* = 3). For the glycoproteomic mass spectrometric assay, the cells were grown in 15 replicate T75 flasks for each group (*n* = 15). After culturing, the established general protocol for all of the mass spectrometric analyses was used [22,24].

#### 4.2.2. Glycomics Assay

The N-Glycan profiling was performed using an Agilent 6200 series nanoHPLC-Chip-QTOF-MS (Agilent) with an Agilent 6210 Time-of-Flight mass spectrometer. The chip (glycan chip II, Agilent) contained a 9 mm × 0.075 mm i.d. enrichment column coupled to a 43 mm × 0.075 mm i.d. analytical column; both are packed with 5-μm porous graphitized carbon (PGC). The N-glycan samples were reconstituted in 40 μL of water, and 5 μL of the resulting solution was used for injection into the LC–MS/MS system. Alongside the samples, blanks and quality controls (N-glycans released from a 1:1 mixture of RNAse B and commercially available serum) were included to ensure data reproducibility and quality. Upon injection, the sample was loaded onto the enrichment column using 3% ACN containing 0.1% formic acid (FA, Fluka, St. Louis, MO, USA). After the analytical column was switched in-line, the nano pump delivered a gradient of 3% ACN with 0.1% FA (solvent A) and 90% ACN with 1% FA (solvent B). The sample was delivered by the capillary pump to the enrichment column at a flow rate of 3 μL/min and separated in the analytical column by the nano pump at a flow rate of 0.3 μL/min using a gradient optimized for N-glycans (0% B, 0–2.5 min; 0 to 16% B, 2.5–20 min; 16 to 44% B, 20–30 min; 44 to 100% B, 30–35 min; and 100% B, 35–45 min) followed by 20-min equilibration for pure A. The tandem MS spectra were acquired via collision-induced dissociation (CID).

Analysis of the N-glycan data was performed using MassHunter Qualitative Analysis Software B.07.00 (Agilent Technologies, Santa Clara, CA, USA). Matching of the monoisotopic masses obtained was completed against our in-house database for glycan composition identification and subsequently verified through their corresponding MS/MS spectra. The relative abundance of each glycan in a sample was determined using the peak area of all glycans from extracted ion chromatograms. The comparison between the relative abundances of primary N-glycan types—high-mannose, undecorated, fucosylated, sialylated, and sialofucosylated glycans—was completed by adding the relative abundances of each glycoform belonging to a specific glycan type. Further comparison of each glycoform was completed using multiple *t*-tests (GraphPad version 9.3.1 for Windows, GraphPad Software, San Diego, CA, USA) at a significance level of α = 0.05. Significantly different N-glycans were mapped on the N-glycan biosynthesis pathway, based on the known biosynthetic sequence [22,23].

#### 4.2.3. Proteomics and Glycoproteomics Assay

The pellets containing membrane proteins were reconstituted with 60 µL of 8 M urea and sonicated for 20 min for denaturation. Two microliters (2 µL) Dithiothreitol (DTT, 550 mM in 50 mM NH_4_HCO_3_) were then added to the samples and incubated for 50 min at 55 °C. The free cysteine was alkylated with 4 µL of iodoacetamide (450 mM) for 20 min in the dark at an ambient temperature. To quench the reaction, 420 µL of a protein digestion buffer (50 mM NH_4_HCO_3_) was added. Trypsin (10 µL, 0.1 mg/mL) was then added to the mixture and tryptic digestion was performed at 37 °C for 18 h. The resulting peptides were purified using a C18 solid-phase extraction cartridge and dried before LC–MS/MS analysis. To enrich the glycopeptides, the tryptic digests were cleaned up using HILIC solid-phase extraction and dried before LC–MS/MS analysis. The purified peptides were adjusted to 0.5 µg/µL while the glycopeptides were adjusted to 0.2 µg/µL before injection using Pierce BCA assay kit, following the manufacturer’s instructions (ThermoFisher, Waltham, MA, USA).

The proteomics and glycoproteomics samples were characterized using an UltiMate™ WPS-3000RS nanoLC system coupled with an Orbitrap Fusion Lumos MS system (ThermoFisher Scientific). One (1) µL of each sample was injected, alongside blanks and quality controls (tryptic digest of commercially available HeLa proteins), which were included to ensure data reproducibility and quality. The analytes were separated using an Acclaim™ PepMap™ 100 C18 LC Column (3 mm, 0.075 mm × 250 mm, ThermoFisher Scientific) at a flow rate of 300 nL/min. Water containing 0.1% formic acid and 80% acetonitrile containing 0.1% formic acid were used as solvents A and B, respectively. The MS spectra were collected with a mass range of *m*/*z* 600–2000 at a rate of 1.5 s per spectrum in positive ionization mode. The filtered precursor ions in each MS spectrum were subjected to fragmentation through 30% higher-energy C-trap dissociation (HCD) using nitrogen gas as carrier.

The mass spectrometry data were analyzed using Byos workflow (Protein Metrics, Cupertino, CA, USA). For the qualitative analysis in Byonic (version 4.3.4, Protein Metrics, Cupertino, CA, USA), the proteins were identified against the human proteome database [59] using a precursor mass tolerance of 20 ppm and fragment mass tolerance of 10 ppm. The digestion parameters used included C-terminal cleavage by trypsin (K and R cleavage sites), with at most two missed cleavages. The following peptide modifications were included: carbamidomethyl @ C; oxidation @ M; deamidation @ N and Q; acetylation at protein N-terminal; Gln to pyro-Glu at N-terminal Q; Glu to pyro-Glu at N-terminal E. The protein IDs were filtered at 1% FDR. To identify the glycoproteins and glycoforms, an additional search was performed in Byonic, using an in-house N-glycan database. Quantification for each protein was completed in Byologic (version 4.3-117, Protein Metrics, Cupertino, CA, USA) by quantifying the XIC area sum of the top three most abundant peptides. The XICs were then normalized to sum total before statistical analysis. On the other hand, glycoform quantification was normalized to each protein’s glycosite to yield the percentage occupancy of a particular glycoform.

#### 4.2.4. Gene Ontology Analysis

To identify significantly different proteins and glycopeptides, multiple *t*-tests were conducted in GraphPad Prism (version 9.3.1 for Windows, GraphPad Software, San Diego, CA, USA), using an FDR approach (FDR = 5%). Significantly over- and under-expressed protein IDs were annotated, using Gene Set Enrichment Analysis [60]. Similarly, the glycopeptides were annotated using g: Profiler [61] to yield significantly enriched pathways and then plotted as a heatmap in GraphPad Prism.

### 4.3. Computational Methods

#### 4.3.1. Network Pharmacology

A ligand database was prepared by downloading the structure data files (.sdf) from several online databases, such as the Comparative Toxicogenomics database (http://ctdbase.org/, accessed on 21 August 2021) [43], STITCH database (http://stitch.embl.de/, accessed on 21 August 2021) [62], GeneCards (https://www.genecards.org/, accessed on 21 August 2021) [63], the Drug Gene Interaction database (http://www.dgidb.org/, accessed on 21 August 2021) [64], and the Protein Databank (https://www.rcsb.org/, accessed on 21 August 2021) [65], and 185 compounds predicted to bind or interact with glycosylation enzymes from the DrugBank (https://go.drugbank.com/, accessed on 21 August 2021) [66]. The drug–gene interactions were predicted using the STITCH database and the Comparative Toxicogenomics database and then visualized using Cytoscape [67].

#### 4.3.2. In Silico Docking

All compounds from the ligand database were loaded onto PyRx [68] and minimized using the Universal Force Field [69], as implemented in Open Babel [70]. The enzymes GlcNAcT-V, Alpha2,6-ST I, and Alpha1-6FucT were selected as the drug target for this study due to the availability of their 3D crystal structures in PDB. The enzyme GlcNAcT-V (PDB ID: 5ZIC, 2.10 Å) [71] was downloaded as a complex with its acceptor sugar, 2-acetamido-2-deoxy-beta-D-glucopyranose-(1-2)-6-thio-alpha-D-mannopyranose-(1-6)-beta-d-mannopyranose. The Alpha2,6-ST I (PDB ID: 4JS2, 2.30 Å) [46] was downloaded as a complex with cytidine monophosphate. The human Alpha1-6FucT (PDB ID: 2de0) [72] was homology-modeled from *Caenorhabditis elegans* POFUT1 (PDB ID: 3ZY6, 1.91 Å) [73] in complex with GDP-fucose, using SWISS-MODELLER [74]. These protein structures were prepared for docking by using the Dock Prep protocol in Chimera [75]. The prepared protein structures were loaded in PyRx as macromolecule receptors.

In silico screening methods were performed in PyRx [68] using the AutoDock VINA docking protocol [76], at exhaustiveness level 8. Validation of the docking protocol was completed by redocking the ligands, complexed with their respective enzymes, using precise grid box parameters. After docking validation, all of the compounds in the ligand database were screened against each of the three enzymes. The compounds were ranked according to VINA-predicted binding energy (kcal/mol). The top binding molecules against each enzyme were visualized for residue interactions with the target enzyme, using Discovery Studio™ (version 21.1.0.20298, Dassault Systemes, BIOVIA Corp, San Diego, CA, USA). From the screening, pictilisib was found to have high compound cross-reactivity (binding to multiple enzyme targets) and a high number of network interactions, and was selected for further in vitro studies.

#### 4.3.3. Glycoprotein Molecular Modeling and Molecular Dynamics

The top glycoproteins were modeled to visualize the effects of changes in glycosylation on protein dynamics and interactions. Interesting glycoproteins, such as ITGA5 and ITGB1, were modeled for visualization of the changes in glycosylation to protein dynamics and interactions. The glycoforms in each glycosite were selected, based on the highest fold-change between the pictilisib- and negative control-treated cells from the glycoproteomics results. The crystal structures of selected proteins were downloaded from PDB, and then the glycans were attached to these proteins using CHARMM-GUI Glycan modeler [77]. The system was solvated using the TIP3P model, and 150 mM KCl was added. The CHARMM36 force field was used for both proteins and carbohydrates [78]. The resulting molecular dynamics input files were used to simulate the glycoprotein dynamics for one ns (10,000 fs/timestep) using the DOST-ASTI High-Performance Computing (HPC; Cluster, Quezon City, Philippines). Simulations were visualized, and the number of interacting hydrogen bonds between the glycans and proteins was analyzed using VMD [79].

## 5. Conclusions

This study highlights the use of combined mass spectrometric (glycomics, proteomics, and glycoproteomics) and bioinformatics approaches that resulted in the identification of an inhibitor, pictilisib, affecting protein N-glycosylation and the associated glycoprotein pathways. By integrating network pharmacology and in silico docking approaches, we were able to identify a glycosylation inhibitor, and subsequently utilized integrated glycomic-driven glycoproteomic and proteomic mass spectrometric analyses to validate its effect in lung cancer cells. The compound was validated to inhibit the formation of fucosylated and sialylated N-glycans attached to glycoproteins involved in apoptosis, cell adhesion, DNA damage repair, and chemical response processes. Furthermore, the compound also significantly affected the cellular processes involved in cell cycle, apoptosis, cell adhesion, transcription, and translation, which we validated using in vitro biochemical assays. Finally, we simulated the differences in interactions of a model glycoprotein complex, ITGA5-ITGB1, due to glycosylation alteration after pictilisib treatment of A549 cells. To further validate the efficacy of the compound in affecting protein glycosylation, it is recommended to conduct future in vivo studies with animal models, coupled with our glycomics-based glycoproteomics mass spectrometric method. These combined methods may be used as tools to reveal potential untapped glycosylation inhibitors.

## Figures and Tables

**Figure 1 molecules-27-03834-f001:**
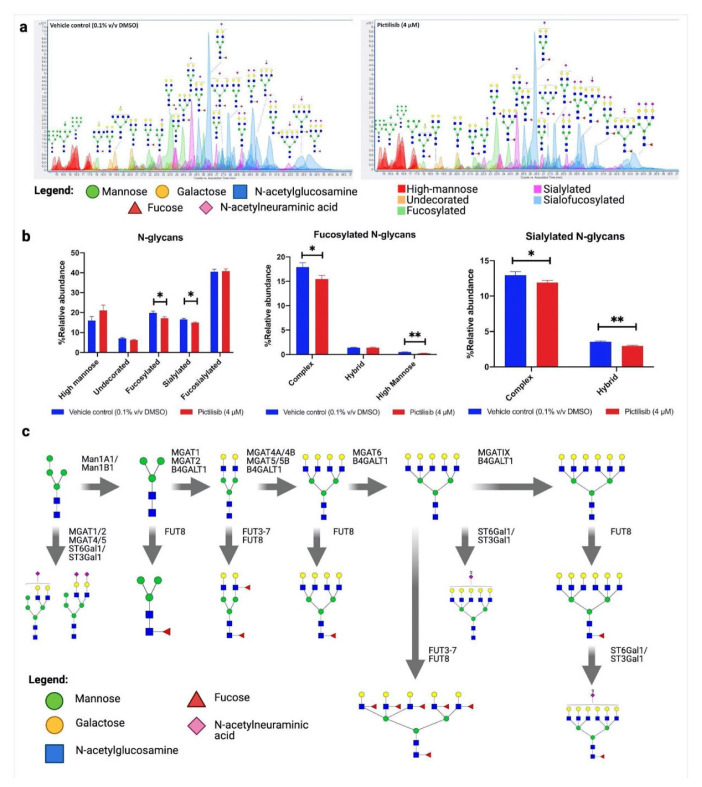
Pictilisib significantly reduced the relative abundance of fucosylated and sialylated N-glycans. (**a**) Glycan-annotated extracted ion chromatograms (EICs) of N-glycomes of vehicle control and pictilisib-treated A549 cells; (**b**) Relative abundances of N-glycan types in vehicle control and pictilisib-treated A549 cells; (**c**) Biosynthetic map showing the abundance of each significantly different N-glycan (Table 1), N-glycan precursor, and known enzymes catalyzing the glycosylation reaction. The data represent mean ± SD; *n* = 3. The statistical differences were detected utilizing multiple *t*-test with FDR correction, * *q* < 0.05; ** *q* < 0.01.

**Figure 2 molecules-27-03834-f002:**
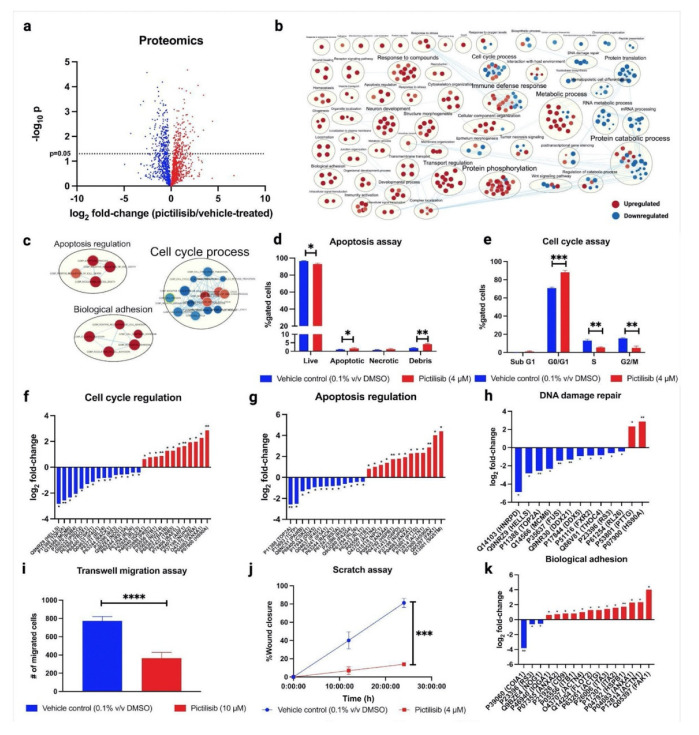
Pictilisib treatment significantly affected the pathways involving ECM interactions and migration, and cell death and proliferation, in A549. (**a**) Volcano plot of differentially expressed proteins in pictilisib-treated A549 cells; (**b**) Gene-set enrichment analysis of pre-ranked protein expression profiles of pictilisib- vs. vehicle control-treated cells; (**c**) Processes involved in apoptosis regulation and biological adhesion were upregulated, while processes involved in cell cycle regulation were downregulated; (**d**,**e**) In vitro assays of pictilisib-treated cells show significantly increased apoptosis and G0/G1 cell cycle arrest; (**f**–**h**) Quantification of proteins related to cell cycle regulation, apoptosis regulation, and DNA damage repair show significant differences (*q*-value < 0.05); (**i**,**j**) In vitro scratch and transwell migration assays show significantly reduced migration activity of pictilisib-treated cells; (**k**) Quantification of proteins related to biological adhesion shows significant differences (*q*-value < 0.05). The data represent mean ± SD; *n* = 3. The statistical differences were detected utilizing multiple *t*-test with FDR correction, * *q* < 0.05; ** *q* < 0.01; *** *q <* 0.001; **** *q* < 0.0001.

**Figure 3 molecules-27-03834-f003:**
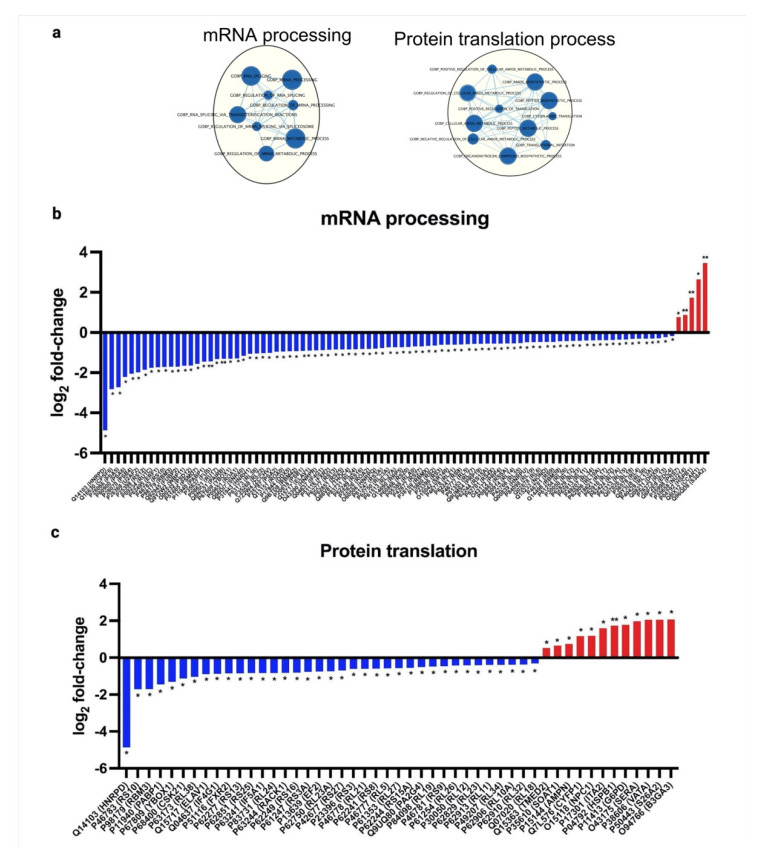
Pictilisib treatment also significantly affected pathways involving mRNA processing and protein translation in A549. (**a**) Based on the GSEA analysis, clusters of pathways involved in mRNA processing and protein translation are downregulated. Most genes involved in mRNA processing (**b**) and protein translation (**c**) are significantly underexpressed following pictilisib treatment. The data represent mean ± SD; *n* = 3. The statistical differences were detected utilizing multiple *t*-test with FDR correction, * *q* < 0.05; ** *q* < 0.01.

**Figure 4 molecules-27-03834-f004:**
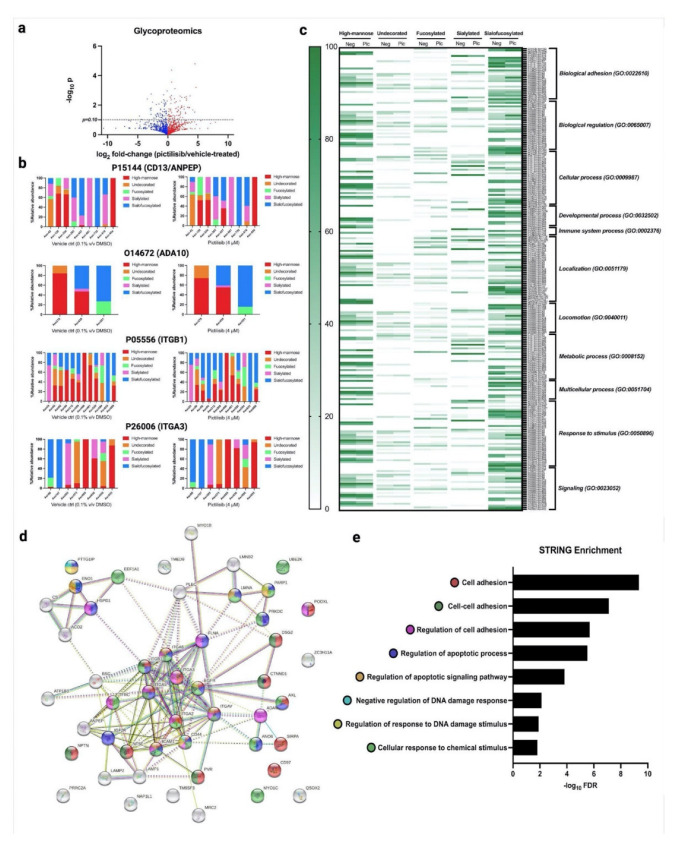
Pictilisib treatment reduced sialylation and fucosylation in specific glycoproteins. (**a**) Volcano plot of differentially abundant glycopeptides; (**b**) Site-specific glycosylation—high-mannose, undecorated, fucosylated, sialylated, and sialofucosylated—of several glycoproteins shown to have reduced fucosylation, sialylation, or sialofucosylation following pictilisib treatment; (**c**) Gene ontology analysis of proteins with reduced fucosylation, sialylation, and fucosylation shows glycoproteins involved in several biological processes; (**d**) STRING interaction analysis shows the interaction of the glycoproteins with reduced fucosylation, sialylation, or sialofucosylation; (**e**) Subsequent STRING enrichment analysis shows a significant enrichment of biological processes involved in adhesion, apoptosis, response to chemicals, and DNA damage. The data represent mean ± SD; *n* = 3. The statistical differences were detected utilizing multiple *t*-test with FDR correction.

**Figure 5 molecules-27-03834-f005:**
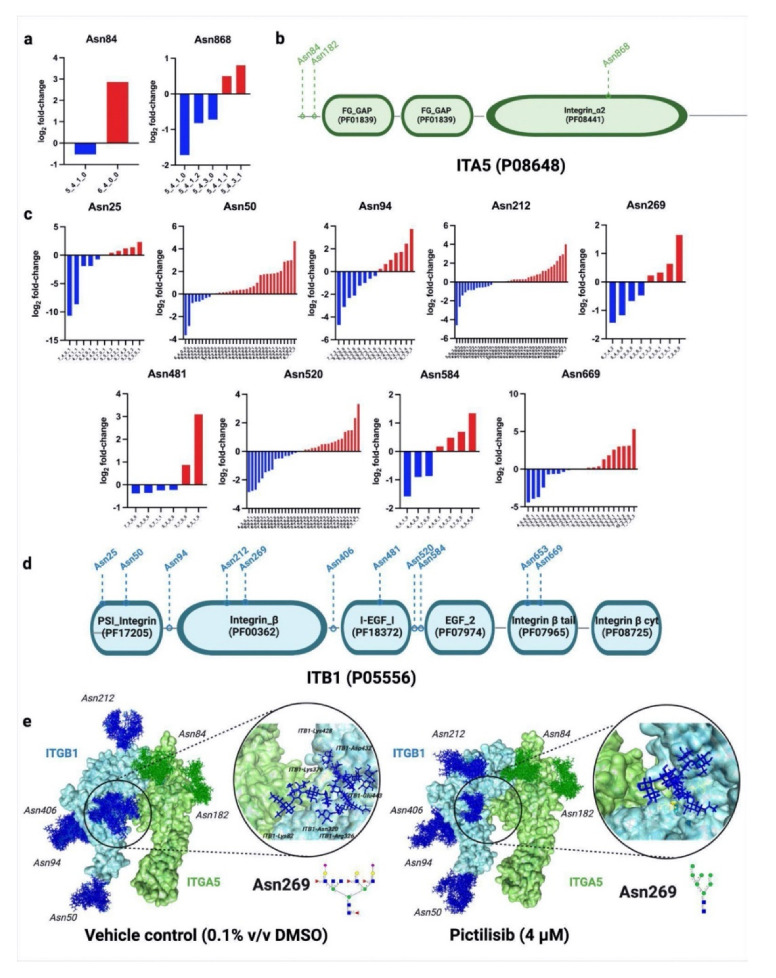
Site-specific glycosylation analysis of ITGA5-ITGB1 illustrates how specific glycosites potentially contribute to protein interactions. (**a**) Site-specific glycosylation of ITGA5 glycoprotein. Bar graphs represent log2 fold-changes in glycoform abundance following pictilisib treatment. Glycosite Asn182 did not change glycosylation; (**b**) Site-specific glycosylation overlaid with protein domain information of ITGA5, annotated using PFAM (http://pfam.xfam.org/, accessed on 15 May 2021); (**c**) Site-specific glycosylation of ITGB1 glycoprotein. Bar graphs represent log2 fold-changes in glycoform abundance following pictilisib treatment. Glycosites Asn406 and Asn653 did not change glycosylation. The *X*-axis represents glycoforms, annotated as Hex_a_HexNAc_b_Fuc_c_Sia_d_; (**d**) Site-specific glycosylation overlaid with protein domain information of ITGB1, annotated using PFAM (http://pfam.xfam.org/, accessed on 15 May 2021); (**e**) 3D trajectories of ITGA5-ITGB1 glycoprotein complexes following treatment with pictilisib. Specific glycoform structures can be seen in Appendix A. Dynamics simulation of negative control (Appendix A) and pictilisib-treated (Appendix A) are also available.

**Figure 6 molecules-27-03834-f006:**
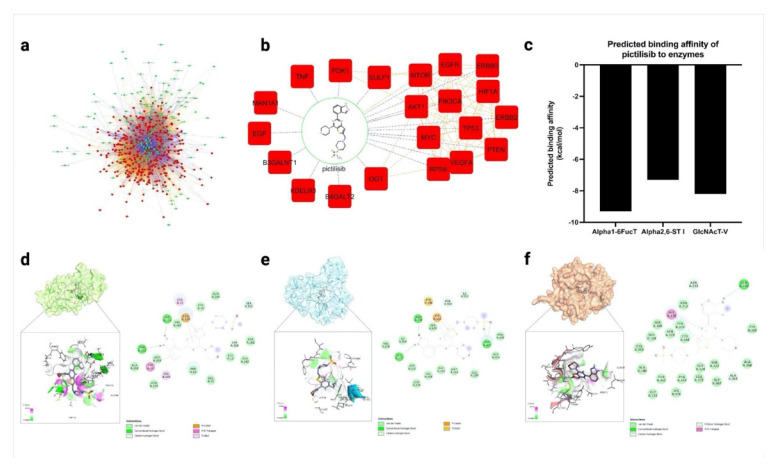
Pictilisib was predicted to interact and inhibit several glycosylation-related genes through network pharmacology and in silico binding approach. (**a**) Drug–gene interaction network of selected glycosylation targeting compounds; (**b**) Sub-network of pictilisib drug–gene interactions; (**c**) The binding affinity of pictilisib against Alpha1-6FucT, GlcNAcT-V, and Alpha2,6-ST I; (**d**–**f**) Docking conformation and residue interactions of pictilisib with Alpha1-6FucT, GlcNAcT-V, and Alpha2,6-ST I, respectively.

**Table 1 molecules-27-03834-t001:** Glycan composition, putative structures, and log 2 fold-change (and *q*-values) of highly significantly different glycans in pictilisib-treated A549 cells compared to vehicle control.

Glycan Composition	Putative Structure	log_2_ Fold-Change	−log_10_ *q*-Value
Hex_8_HexNAc_7_NeuAc_2_	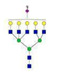	−2.4214	1.6249
Hex_8_HexNAc_7_Fuc_6_	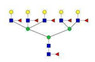	−1.6756	1.6249
Hex_9_HexNAc_8_Fuc_1_NeuAc_2_	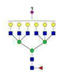	−1.3536	1.5544
Hex_9_HexNAc_8_Fuc_1_	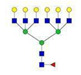	−1.3269	1.5403
Hex_6_HexNAc_4_NeuAc_2_	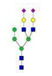	−1.0587	1.6249
Hex_3_HexNAc_2_Fuc_1_	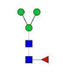	−1.0000	1.4854
Hex_5_HexNAc_4_Fuc_2_	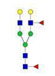	−0.7675	1.6249
Hex_6_HexNAc_4_NeuAc_1_	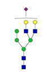	−0.5385	1.6249
Hex_7_HexNAc_6_Fuc_1_	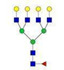	−0.3078	1.6249

## Data Availability

The data presented in this study are available on request from the corresponding author.

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
