# Peer review of "An Integrated Mass Spectrometry-Based Glycomics-Driven Glycoproteomics Analytical Platform to Functionally Characterize Glycosylation Inhibitors"

_molecules, 2022, doi:10.3390/molecules27123834_

Round 1
Reviewer 1 Report
Please clarify the sentence at line 449 "To quench the reaction, 420 buffer " , unit is missing, or?
Reviewer 2 Report
This study performed an integrated bioinformatics and mass spectrometry-based glycomics-driven glycoproteomics analysis pipeline to identify an N-glycan inhibitor against lung cancer cells. And combined network pharmacology and in-silico screening approaches to identify a potential inhibitor, pictilisib, against several glycosylation-related proteins such as Al- pha1-6FucT, GlcNAcT-V, and Alpha2,6-ST-I. However, in its current state, the paper showed some deficiencies. The paper is not innovative enough. Moreover, the discussion of the results was insufficient and lacked depth.
- This study is limited to theoretical verification, and animal experiments should also be carried out to clarify its mechanism of action.
- Possible drawbacks and limitations of the study should be indicated.
Reviewer 3 Report
Figure 1:
The observed glycan changes are very small. Its possible the individual glycans might change more, but are so low in abundance that they are overshadowed by non-changing glycans, eg. when looking at total fucosylation and sialylation. In the supp fig 1. the high mannose change seems the most relevant, however I do not see replicates or error bars so I would not conclude anything at this point.
Pictilisib is a PI3K inhbitor and known for its anti-neoplastic properties. The authors are attempting to link its anti cancer properties to glycan changes. Assuming these changes are relevant, the authors must run the same experiments on different drugs known to modulate glycans as a positive control and on another drug from the list they provided, to see if their conclusion stands or not. For all we know, the effect on the proteome could be specific to this drug and not because of its possible glycan modulatory properties.
The conclusion on ITGA5-ITGB1 interaction needs to be supported by in vitro and in cellulo experiments. Native blue PAGE can be used to determine oligomerization. Immunoprecipitation is another method. Staining for both molecules in the cell and analyzing their co-localization or a pulse chase experiment should be performed to validate the in silico data.
It is worth noting that many layers of regulation govern the glycan output, not just glyco-enzyme overexpression. Other factors include glycosyltransferases interactions and complex formation with nucleotide sugar transporters (see Sakari Kellokumpu studies), glycosyltransferase retention by GOLPH3 (see Ricardo Rizzo and Alberto Luini studies), golgi structure and hypoxia/redox . These factors should be included in the introduction and discussion and should be considered when testing these drugs on the cells.
Round 2
Reviewer 3 Report
Authors clarified most of the points and stated some of the shortcomings in the manuscript. Although it can still be improved, it can be published in its present form.